# Rational Design of Effective Binders for LiFePO_4_ Cathodes

**DOI:** 10.3390/polym13183146

**Published:** 2021-09-17

**Authors:** Shu Huang, Xiaoting Huang, Youyuan Huang, Xueqin He, Haitao Zhuo, Shaojun Chen

**Affiliations:** 1College of Chemistry and Environmental Engineering, Shenzhen University, Shenzhen 518060, China; hnlgnh@126.com; 2College of Physics and Optoelectronic Engineering, Shenzhen University, Shenzhen 518060, China; 3Guangdong Research Center for Interfacial Engineering of Functional Materials, Shenzhen Key Laboratory of Polymer Science and Technology, College of Materials Science and Engineering, Shenzhen University, Shenzhen 518060, China; hxt_tf@sina.cn; 4Shenzhen BTR New Material Group Co., Ltd., High-Tech Industrial Park, Xitian, Gongming Town, Guangming New District, Shenzhen 518106, China; huangyouyuan@btrchina.com (Y.H.); hxcome@btrchina.com (X.H.)

**Keywords:** polymer binder, structure design, electrochemical performance, Li-ion batteries, LiFePO_4_ cathode

## Abstract

Polymer binders are critical auxiliary additives to Li-ion batteries that provide adhesion and cohesion for electrodes to maintain conductive networks upon charge/discharge processes. Therefore, polymer binders become interconnected electrode structures affecting electrochemical performances, especially in LiFePO_4_ cathodes with one-dimensional Li^+^ channels. In this paper, recent improvements in the polymer binders used in the LiFePO_4_ cathodes of Li-ion batteries are reviewed in terms of structural design, synthetic methods, and working mechanisms. The polymer binders were classified into three types depending on their effects on the performances of LiFePO_4_ cathodes. The first consisted of PVDF and related composites, and the second relied on waterborne and conductive binders. Profound insights into the ability of binder structures to enhance cathode performance were discovered. Overcoming the bottleneck shortage originating from olivine structure LiFePO_4_ using efficient polymer structures is discussed. We forecast design principles for the polymer binders used in the high-performance LiFePO_4_ cathodes of Li-ion batteries. Finally, perspectives on the application of future binder designs for electrodes with poor conductivity are presented to provide possible design directions for chemical structures.

## 1. Introduction

Lithium-ion batteries (LIBs) are important rechargeable power sources that were first commercialized in 1991 and rapidly proceeded to occupy the markets of rechargeable batteries due to their safety, high power density, and low-cost [1,2,3,4]. LIBs currently are used for widespread applications in electric vehicles (EVs), portable electronic devices, and power grid storage [5,6]. For instance, the demand for LIBs rose at an annual rate of 30% between 2016 and 2020 thanks to the high-speed development of energy storage devices. This motivated a great revolution in the development of electrodes with high volumetric/gravimetric energy density and long-term cycle life [7,8]. Hence, satisfying the ever-increasing demand for LIBs is urgent and vital for many industries as well as society [9,10,11]. Materials comprising the major components of electrodes are tightly linked to the demand and the market prospects for LIBs [12,13,14,15,16,17]. Meanwhile, electrode structures are key parameters affecting the overall electrochemical performances of LIBs [18]. In this respect, the structure of LiFePO_4_ (LFP) is advantageous in terms of its low cost, environmental friendliness, outstanding thermal stability, and safety. Hence, it became one of the first industrialized electrode materials for LIBs [19,20,21,22]. However, LFP cathodes still suffer from several intrinsic drawbacks such as poor conductivity and low Li^+^ diffusion due to their one-dimensional Li^+^ channels, and thereby are extremely limiting to high rates of application [23,24,25]. As a result, increasing attention has been paid to constructing novel structures of active materials wherein polymer binders play vital roles in electrochemical performances due to their unique and efficient chemical structures despite limited mass content [26,27,28,29,30].

Currently, polymer binders are no longer regarded only as merely bond reagents but also as contributors to the conductivity and the solid electrolyte interphase (SEI) stabilization of anodes/cathodes [31,32]. Polymer binders are inactive components in electrodes but are as vital as the active materials and the other ingredients [33]. Outstanding binders should provide powerful binding affinities between the active material particles and the current collectors to maintain electrode integrity, even under limited ion diffusion and side reactions. In the process of slurry preparation, polymer chains coat LFP particle surfaces upon stirring. Next, the produced slurry is coated onto aluminum foils, and electrodes are created upon drying. Therefore, the polymer structure coated onto the LFP particle surface significantly affects the performance of electrodes. Efficient polymer structures could promote the diffusion of Li^+^ to a degree, which would be helpful for mitigating the shortage of one-dimensional Li^+^ channels in LFP materials. Consequently, binders of LIBs have been increasingly investigated over the years to provide novel concepts for designing new polymer structures. As a result, polymer binders have become strategic components for alleviating the cyclability problems of batteries. In this regard, an ideal electrode model should be able to fulfill several aspects. The first consists of maintaining enough adhesion to prevent delamination and separation of electrodes during charge/discharge processes. The second relies on compatibility with slurry preparation and electrode fabrication in addition to offering continuous conductive networks within electrodes. This also facilitates the formation of electron and ion circuits to maintain the transfer of Li^+^ and guarantee effective electrochemical reactions. On the other hand, providing electrochemical, chemical, and thermal stability under severe internal and external battery environments is also important. Finally, a low cost of widescale commercial application is highly desirable. For LFP materials, three major strategies have been utilized for binder development. The first consists of a traditional polyvinylidene fluoride (PVDF) binder used for commercial applications. The second consists of waterborne binders with eco–environmental properties. The third relies on functional binders such as the conductive or ionomer polymers used to overcome shortages of LFP structures. The use of efficient binders in LFP cathodes could enhance conductivity and promote ion diffusion, thereby improving their properties and long-term cycle life.

Given the important role of binders in functional LIBs, summarizing their rational design and investigating their functional chemical structures and the working mechanisms of binders for LFP cathodes has become vital for research and development. Elaborating and evaluating the various polymer binders for LFP materials are also crucial. Therefore, the design principles of polymer chains at the molecular level and their effects on the electrochemical characteristics of LFP cathodes are reviewed in this paper. The binders were divided according to their molecular structures, and binder strategies were elucidated based on their functionalities. Based on the evolution of binder research, three types were categorized: (i) PVDF and related composites; (ii) waterborne binders; and (iii) conductive binders. Unlike other recent reviews on polymer binders, this review focuses on the design and selection of functional groups as well as on the tailoring of polymer chains using the various design strategies influencing LFP materials. In addition, the mechanisms of binder structures for promoting Li-ion diffusion in LFP materials are highlighted to clarify the interactions and relationships among polymer structures, material particle surfaces, and electrochemical performance. Finally, recent progress is summarized to forecast future binder research directions as well as to predict the development tendencies of the binder systems of the LFP cathodes of LIBs and other rechargeable batteries.

## 2. Functions and Effects of Binders and Their Rational Design

### 2.1. PVDF and Related Composites

Poly(vinylidene fluoride) (PVDF) is a traditional commercial binder generally applied in LFP cathodes of LIBs thanks to its good electrochemical stability, relevant electrolyte absorption, and adhesion ability [34,35,36,37]. However, PVDF must be dissolved in strongly polarized N-methyl pyrrolidone (NMP) solvent, which is relatively toxic, costly, and environmentally unfriendly [38]. For wider applications, PVDF cannot effectively enhance the performances of LFP cathodes because it possesses weak van der Waals forces that are unable to provide durably strong adhesion for active materials during cycling. In addition, PVDF is not good for enhancing the conductivity and ion diffusion of LFP cathodes due to its restricted molecular structure. Therefore, tremendous efforts have been devoted to decorating polymer chains to yield related PVDF-based complex polymers. For instance, Hu et al. [39] used poly(vinylidenefluoride-co-hexa-fluoropropylene) (PVDF-HFP) binder in LFP cathodes to yield good electrochemical performances with enhanced cycling performances (Figure 1a). EIS tests show that the PVDF-HFP/LFP cathode present lower resistance demonstrating a faster ionic transport in polyether-based electrolytes (Figure 1b,c). These outcomes are associated with the presence of a flexible, amorphous and electronic conductivity coupled with a lower glass transition temperature (Tg). The amorphous domains of the polymer chains trapped many liquid electrolytes, and the crystalline regions provided enough mechanical strength. Other studies have shown poly(vinylidene fluoride-co-trifluoroethylene) (PVDF-TrFE) binder [40] to yield LFP cathodes with better rate capabilities and cycle performances than PVDF-HFP cathodes. Thus, the amorphous phase content and the Tg of polymers can affect Li^+^ diffusion. Zhang et al. [41] used polyimide (PI) to replace part of PVDF, and the LFP-PI-0.7 (mixture of 30 wt.% PVDF and 70 wt.% PI as binders) electrodes yielded significant cycling stability over 300 cycles at 1C with a stabilized capacity of 141 mA h g^−1^. Moreover, the capacity retentions of LFP-PI electrodes were promoted with the increase in PI contents. These outcomes were due to the high ion conductivity of the PI binder, which is 3.6-fold higher than that of PVDF. The PI binder did not only maintain stable distribution between LFP particles to effectively restrain the re-aggregation of LFP particles upon repeated cycling processes but also boosted ion diffusion through electrodes. Furthermore, the active carbonyl groups in the PI skeleton may have yielded extra capacity due to the reversible enolization reaction. However, the usage of highly toxic organic solvents such as NMP is not desirable. Therefore, waterborne, fluorine-containing polymer polytetrafluoroethylene (PTFE) has been used to replace organic PVDF in LiFePO_4_/C cathodes (Figure 2a,b) [42]. Measurements have revealed PTFE cathodes to possess higher capacities (161.1 mAh g^−1^) than PVDF cathodes (150.7 mAh g^−1^) (Figure 2c,d). Therefore, PTFE can be utilized as a binder to replace conventional PVDF binders for waterborne-processed LFP cathodes in LIBs. As a result, the design of complex and multifunctional polymer systems can break the bottleneck structure of LFP electrodes, offering novel ideas for further research. However, most PVDF-based binders still require the use of organic solvents and show no enhanced ion diffusion in electrodes to further increase cycling performance.

### 2.2. Waterborne Binders

The transition of LFP materials through aqueous processes must be achieved in order to realize eco–environmental and low-cost development and applications. To this end, Porcher et al. [43,44,45,46] explored the stability of LiFePO_4_ materials in water. Their results showed that the contact of LFP with water yielded a few nanometers-thick Li_3_PO_4_ layer at the grain surface. Further research revealed slight changes in LFP particles upon aging, which did not affect the electrochemical behavior. However, the Fe_solid_^III^ content in active materials increased, thereby decreasing the energy density of LFP electrodes. Consequently, waterborne processes should satisfy several requirements. The first requirement is limited immersion time. The second involves the solid content of slurries, which should be high. The third involves natural pH, which should not be modified so as not to restrain the accelerated dissolution of active materials. Finally, the active material should be stored in a dry atmosphere.

Novel binder systems that are soluble in water and compatible with cathodes have also attracted increasing attention. For instance, a novel, water-soluble elastomer binder (WSB) [47] was added to carboxymethyl cellulose (CMC) to enhance the flexibility and adhesion of binders in elastomer-based LiFePO_4_ cathodes. Improved flexibility was beneficial for enhancing the energy density of electrodes. Moreover, the SEM images present uniform electrodes with low-roughness, WSB-coated surface films. When evaluating electrode flexibility, the results show that the elastomer system was much more flexible, about 2-fold more so than the PVdF system. The resulting cathodes showed good performance with capacity reaching 120 mAh g^−1^ at 10C and 60 °C. Furthermore, the addition of poly(acrylic acid) (PAA) to the CMC binder decreased the viscosity of LiFePO_4_ slurries, thereby increasing the solid concentration and enhancing the discharge capacity when compared to CMC alone [48]. In further research, less costly PAA was used alone as a binder in LiFePO_4_ cathodes to reduce the polarization and transportation of Li^+^ during intercalation/deintercalation processes (Figure 3) [49]. The CV test indicated that the PAA electrode showed the better reversibility, ascribing to the lower resistances of solid electrolyte interphase (SEI) on LiFePO_4_ and charge transfer for lithium ion intercalation and de-intercalation (Figure 3a). The PAA/LFP cathode presented a better cyclic stability with a retention of 98.8% after 50 cycles compared to the PVDF one with only about 94.9% (Figure 3b). Moreover, the PAA/LFP cathode maintained integrity compared to the PVDF one with obvious cracks (Figure 3c,d). For example, Lux et al. [50] introduced a CMC aqueous binder on its own that yielded LFP cathodes with an initial capacity of 140 mAh g^−1^ and capacity fading of only 0.025% per cycle. Sun et al. [51] used a water-soluble chitosan derivative (carboxymethyl chitosan (C-CTS)) to investigate CMC-relevant structures. The obtained LFP electrodes were more resistant to the electrolyte and showed higher-rate performances than those prepared with PVDF and CMC. Furthermore, introducing the cyanoethyl group into C-CTS chains to prepare cyanoethylated carboxymethyl chitosan (CN-CCTS) [52] polymers yielded highly adhesive binders. The adhesion strength of CN-CCTS/LFP cathodes was 3.6-fold higher than that of CMC. Cyclic voltammetry (CV) and electrochemical impedance spectroscopy (EIS) results revealed CN-CCTS/LFP cathodes with more favorable electrochemical kinetics than those prepared with CMC and PVDF, thereby having better cycling performances. After 100 cycles, the SEM images of CN-CCTS/LFP cathode show no obvious morphological changes or cracks. The results demonstrate that the waterborne CN-CCTS binder could accommodate the stress/strain and maintain structural/mechanical stability during cycling. To enhance adhesion, AN was employed to modify polyethylenimine (PEI) and synthesize N-cyanoethyl polyethylenimine (CN-PEI) [53]. The introduction of polar cyano groups exhibited outstanding adhesion strength as well as enhanced ionic conductivity. As a result, LFP cathodes containing CN-PEI binders presented enhanced cycling and rate performances with capacity retention reaching 99.6% at a rate of 0.5C after 100 cycles as well as a high capacity of 102.4 mAh g^−1^ at 5C. However, AN is a highly toxic organic chemical despite being an efficient group, limiting the further commercial application of related AN-modified polymers. Hence, further studies should focus on the biopolymers used in Si-based anodes of LIBs to prepare waterborne binders with high flexibility and adhesion [54,55,56,57,58]. These biopolymers have multidimensional networks and hyperbranched structures containing many –OH, –COOH, and –NH_2_ groups in their polymeric chains [59]. Such functional groups formed strong hydrogen bonds with oxygen-containing groups on the surfaces of active material particles and the oxide layers of current collectors [60,61]. In turn, their high adhesion and special 3D structures effectively maintained the integrity of electrodes without separation or cracks upon cycling [62,63,64]. Therefore, biopolymers are promising alternatives for LFP cathodes. In addition to waterborne processing and adhesion problems, Li^+^ transport and the conductivity of LFP cathodes are also bottleneck issues restricting optimized performance.

### 2.3. Conductive Binders

To overcome the bottleneck drawbacks of LFP structures, ionomers and conductive polymers have been explored thanks to their beneficial structures. In this respect, lithiated ionomers have been the focus of increasing attention. For instance, Shi et al. [65] synthesized lithiated poly(perfluoroalkylsulfonyl)imide ionene (PFSILi) followed by blending with PVDF to form a binder for LFP cathodes. At 4C, the capacity and energy density of PFSILi-PVDF/LiFePO_4_ cathodes were 1.5- and 1.66-fold higher than those of nonionic PVDF, respectively. The anionic polymer chains of PFSILi could be fixed and spread across the liquid electrolyte phase, the SEI film, and the LiFePO_4_ solid phase. Hence, PFSILi benefited from the transport of Li^+^ between the interfaces of electrodes and electrolytes due to its high ionic conductivity. This meant that the PFSILi binder could overcome the bottlenecks of LFP cathode development. Given the excellent function of lithiated ionomers, Qiu et al. [66,67,68,69] used cotton raw material to prepare a novel, cellulose-derived lithium carboxymethyl cellulose (CMC-Li) binder (Figure 4a). The obtained LFP cathodes exhibited a capacity of 175 mA h g^−1^ after 200 cycles with little capacity loss (Figure 4b) and obviously enhanced rate performances (Figure 4c). Terpene resin (TS) emulsion was added into lithium polyacrylate (PAALi) to enhance its flexibility and adhesion capabilities [70], providing insight into elaborate designs for blended composite binders in LIBs. In this review, our group designed and prepared a novel, waterborne, lithiated ionomer (PSBA-Li) binder [71] with a limited addition of 1.5 wt.% to yield LFP cathodes with excellent cycling performance retentions of 108.5%, 103.6%, and 114.9% at 0.5C, 1C, and 2C, respectively. Consequently, lithiated ionomers could efficiently enhance the electrochemical properties of LFP cathodes, especially their cycling performances. This can be explained by the uniformly distributed lithiated ionomers on the surfaces of LFP material particles. Secondly, lithiated ionomers could provide more Li^+^ in order to increase the conductivity of electrodes and shorten the distance of Li^+^ from the electrolytes and the surfaces of LiFePO_4_ particles. Finally, the fabricated cathodes could exhibit less redox potential differences, smaller electrode polarizations, and faster Li^+^ diffusion rates, leading to excellent electrochemical performances.

To further increase the conductivity of LFP cathodes, conductive polymers have been investigated. Among polymers, poly(3,4-ethylenedioxythiophene):polystyrene sulfonate (PEDOT:PSS) [72,73,74,75,76] with high conductivity was explored as a binder for LFP cathodes. The obtained electrodes displayed outstanding rate performances with a capacity of 126 mAh g^−1^ at 5C as well as excellent cycling stability at 1C with less than 1% decay after 100 cycles [73]. These values substantially exceeded those of LFP cathodes fabricated with conventional compositions. This was associated with the connectivity of conductive PEDOT:PSS binders in the three-dimensional structures that led to improved overall conductivity of the composite electrodes. The PEDOT:PSS also acted as a conduction-promoting agent when added into CCTS [76] and sulfonated poly(phenylene oxide) (SPPO) [77] to yield conductive composite binders. The introduction of PEDOT:PSS could facilitate the formation of homogeneous conductive bridges throughout LFP cathodes. This could also replace some conductive additives, leading to increased-energy-density cathodes. Moreover, a 10 Ah PEDOT:PSS/CCTS-LFP prismatic cell exhibited remarkable cycling performance with a capacity retention of 89.7% at a 1C/2C (charge/discharge) rate over 1000 cycles as well as a higher rate capability than commercial PVDF-LFP.

In addition to high conductivity, improving the adhesion of polymer binders has also been the focus of recent research. In this respect, sodium alginate (SA) was functionalized with 3,4-propylenedioxythiophene-2,5-dicarboxylic acid (ProDOT) (Figure 5a) to form SA-PProDOT polymer chains [78] containing carboxyl, hydroxyl, and ester groups that promoted the chemical bonding between LFP particles. SA and ProDOT molecules may self-assemble through the hydrophilicity of functional groups to yield interfaces for esterification reactions (Figure 5b). In this process, the produced water could be removed from the interface to the hydrophilic phase, moving the reaction equilibrium forward. Without conductive additives, the fabricated cathodes could achieve the theoretical specific capacities of LiFePO_4_ cathodes, estimated at 170 mAh g^−1^ at C/10 and 120 mAh g^−1^ at 1C over 400 cycles (Figure 5c). The attachment of ProDOT groups could enhance the polarity of the polymer as well as its uptake of the electrolyte, thereby promoting ionic conductivity (Figure 5d). On the other hand, 3D, hierarchical, walnut kernel-shaped conductive polymer binder (CPB) (Figure 6) [79] and ionic framework polymer binder (LPPS) [80] could balance bonding strength and conductivity due to their unique chemical structures. The fabricated LFP cathodes could exhibit robust contacts between active materials and provide efficient pathways for Li^+^ transport. In this process, the effective structures would determine the overall performances of LFP cathodes. In fact, LFP cathodes present excellent cycling stability and high Coulombic efficiency, even at high current densities.

The outcomes for LFP cathodes with various binders are summarized in Table 1. It can be seen that waterborne processes for LFP cathodes have been the focus of development carried out in recent research. In addition, the lithiated ionomers and conductive binders used in LFP cathodes have exhibited better cycling performances with relative long-term cycle lives of 200–400 cycles. On the other hand, research dealing with composite binders has increased due to their multi-functionality with positive effects on performance. In addition, PEDOT:PSS has played a vital role in enhancing conductivity, thereby promising to replace conductive additives. Based on recent outcomes, many efforts have been focused on polymer structures and functions. For polymer structures, one approach is to design and prepare polymers with flexible components to form uniform electrodes with low-roughness, binder-coated surface films. Another approach is to decorate polymer chains with functional groups such as carboxyl, hydroxyl, and ester groups, which could promote chemical bonding between LFP particles to enhance adhesion. Another vital approach is to introduce conductive structures. Therefore, the introduction of functional groups mainly plays a role in forming hydrogen bonds or strong interactions between LFP particle surfaces. Another function focuses on promoting the diffusion of Li^+^ to shorten the pathway of Li^+^ to particle surfaces. Efforts could be made to decrease the addition amounts of binders to enhance energy density and conductivity and enhance the transformation of Li^+^ at the interfaces between the electrodes and electrolytes to overcome the shortages of the LFP structure. Therefore, the ultimate goals would be to make up for the intrinsic drawbacks of olivine structure LiFePO_4_ to realize long-term cycle life and excellent rate performances, especially at high rates. Due to their efficient functions, polymer binders within various categories and with various structures have become the research focus in the last five years, further demonstrating particular functions in LFP materials. The addition of binders could be decreased to below 5 wt.% for further applications, which would be beneficial for increasing the energy densities of electrodes. To further enhance the conductivity of polymer binders, ionic liquid-based copolymers hold promise in this regard. For example, ionic liquid-based copolymers are ionic polymers containing charge-balanced anions and cations featuring high ionic conductivity, excellent thermal stability, and wide electrochemical work windows. Hence, they have been broadly used in battery systems [81,82,83] as binders [84] and electrolytes [85] as well as in many other devices [86,87]. Ionic liquid-based polymers with freely moving ions could promote the mobility and transmission of Li^+^ between electrodes, helpful for cramping plentiful Li^+^ within the limited spaces of battery systems. These features may decrease the resistances as well as raise the Coulombic efficiencies and capacities of electrodes. Therefore, ionic liquid structures can solve shortages of LFP materials with insufficient ion transport, thereby benefiting the realization of LFP cathodes with long-term cycle lives. In addition, these binder systems might also be suitable for other olivine materials such as olivine LiNiPO_4_ [88], LiCo_1/3_Mn_1/3_Ni_1/3_PO_4_ [89], and LiCoPO_4_ [90] cathodes through the use of traditional PVDF binders. Moreover, the introduction of chitosan, cross-linked using an electrodeposit method for the Zn–MnO_2_ system with excellent electrochemical properties, would be a promising strategy for further research [91].

## 3. Conclusions and Prospects

Binders act as “bridges” to bond the active material particles and conductive additives adhering to current collectors. Therefore, binders are indispensable for maintaining the integrity of electrodes and forming electron/Li^+^ conductive routes throughout the entire electrode matrix, thereby influencing the performances of battery systems. As a result, tremendous efforts have been devoted to decorating structures of polymer chains. The fundamental design principles could be summarized in several points. The first consisted of a waterborne process that was advantageous in terms of its eco–environmental benefits and low cost as well as it being a non-toxic solvent. The second involved enhancing conductive ability. Highly conductive PEDOT:PSS could replace conductive additives to fabricate cathodes, and the third involved facilitating the transformation of Li^+^. The lithiated ionomer (PSBA-Li) binder could promote the transformation of Li^+^ in electrodes and shorten the distances of Li^+^ from electrolytes to the surfaces of LiFePO_4_ particles, leading to enhanced cycling stability. The fourth point involved strengthening adhesion ability with less added amounts of binders. SA-PProDOT polymer chains with carboxyl, hydroxyl, and ester groups could promote chemical bonding between LFP particles, leading to stronger adhesion. Moreover, a PSBA-Li binder could be used at a limited 1.5 wt.% to fabricate LFP cathodes for the function of –COOH. The fifth relied on enhanced functionalized structures with various functional groups on polymer chains. Based on the elucidated, verified, and exhibited studies in this review, various binders were found to enhance the electrochemical performances of LFP cathodes by alleviating the drawbacks of LFP materials such as poor conductivity and one-dimensional Li^+^ channels that could be realized by rational polymer design. The structures and working mechanisms of synthetic polymer binders were thoroughly elaborated in this review by emphasizing the relationships between polymer structures and the performances of cathodes.

Further evaluation of functional binders in electrodes could be accomplished by carrying out in-depth and comprehensive studies including in-situ analyses, dynamic simulations, and mechanism propositions to provide deeper theoretical guidance for binder design. Furthermore, full cell systems under extreme conditions should be created to simulate application processes, which would be beneficial for estimating the function of binders. Additionally, more attention should be paid to binders with multiple functions on one polymer chain. The main groups could contain: (1) hydrophilic groups to realize waterborne processes; (2) –COOH, –OH, or –NH_2_ groups, which could form hydrogen bonds with LFP particle surfaces with enhanced adhesion; (3) ion conduction of ionic groups to promote the diffusion of Li^+^; and (4) crosslinked structures to maintain the structural stability of polymers upon cycling. Finally, deeper investigations of design strategies and novel preparations for binders should be performed, not only for LFP cathodes but also for other electrodes for various battery systems.

## Figures and Tables

**Figure 1 polymers-13-03146-f001:**
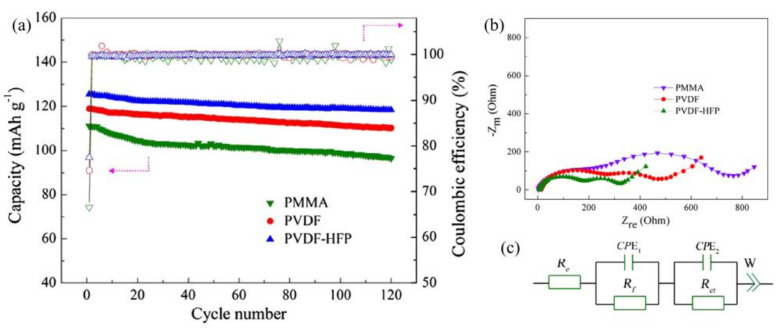
(**a**) Cycling performance and coulombic efficiency of LiFePO_4_/C electrodes with PMMA binder, PVDF binder, and PVDF-HFP binder at room temperature and 1C rate. EIS results of (**b**) the LiFePO_4_/C electrodes with PMMA binder, PVDF binder, and PVDF-HFP binder at frequencies from 0.01 Hz to 100 kHz and (**c**) equivalent circuit. Reprinted with permission from ref. [39]. Copyright 2014 Elsevier.

**Figure 2 polymers-13-03146-f002:**
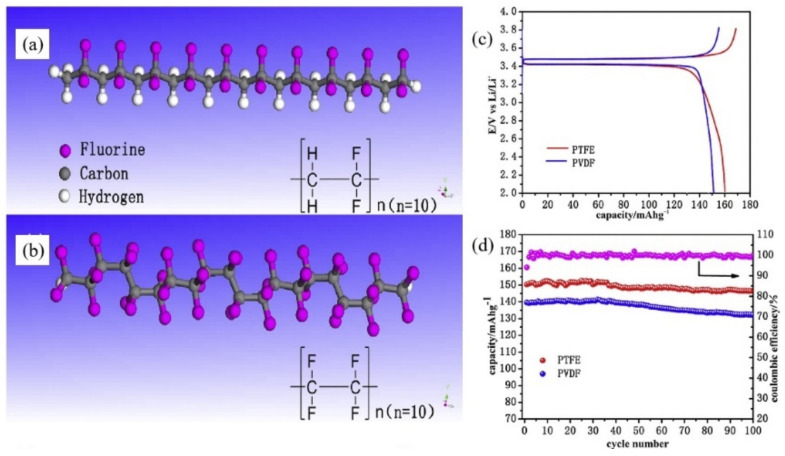
Chemical structures of (**a**) PVDF and (**b**) PTFE. (**c**) The first charge/discharge curves of PTFE and PVDF electrodes at 0.1C. (**d**) Cycle stability of cathodes prepared form PVDF and PTFE evaluated at 0.2C rate. Reprinted with permission from ref. [42]. Copyright 2015 Elsevier.

**Figure 3 polymers-13-03146-f003:**
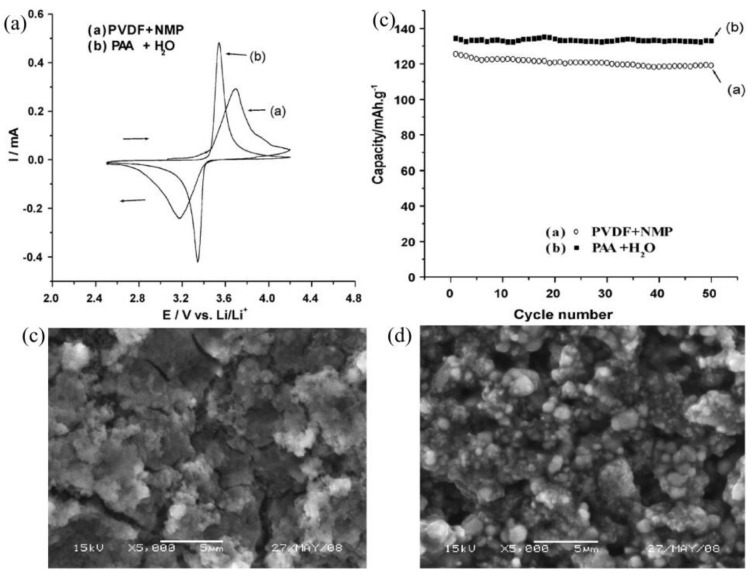
(**a**) Cyclic voltammograms of LiFePO_4_ electrodes in 1-M LiPF_6_/EC:DMC:EMC (volume ratio 1:1:1), scan rate: 0.1 mV s^−1^. SEM images of LiFePO4 electrodes. (**b**) Cyclic stability of LiFePO_4_ electrodes in 1-M LiPF_6_/EC:DMC:EMC (volume ratio = 1:1:1). (**c**) The electrode was prepared in NMP with PVDF as a binder and (**d**) the electrode was prepared in the aqueous solvent with PAA as a binder. Reprinted with permission from ref. [49]. Copyright 2009 Elsevier.

**Figure 4 polymers-13-03146-f004:**
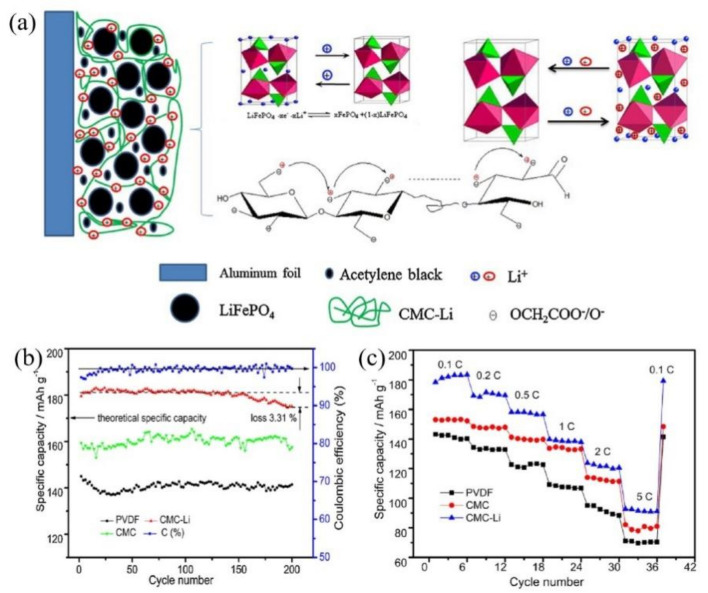
Preliminary investigation of the mechanism of the CMC-Li binder in LiFePO_4_ battery and charge/discharge tests. (**a**) Preliminary investigation of the mechanism; (**b**) Specific capacity; (**c**) Rate performance. Reprinted with permission from ref. [67]. Copyright 2014 Elsevier.

**Figure 5 polymers-13-03146-f005:**
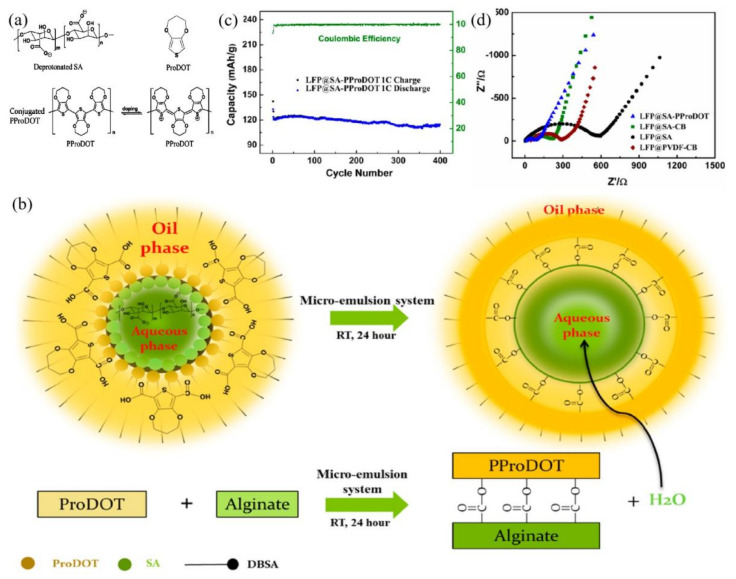
(**a**) Chemical structure of deprotonated SA, ProDOT, and PProDOT. (**b**) Schematic design of the microemulsion system for synthesis of SA-PProDOT polymer. (**c**) Long-term performance of LFP electrodes binder based on SA-PProDOT (LFP:binder = 8:2) binder. (**d**) EIS spectra of the LFP@SA-PProDOT, LFP@SA-CB, LFP@SA, and LFP@PVDF-CB samples between 0.1 Hz and 100 kHz. Reprinted with permission from ref. [78]. Copyright 2015 the American Chemical Society.

**Figure 6 polymers-13-03146-f006:**
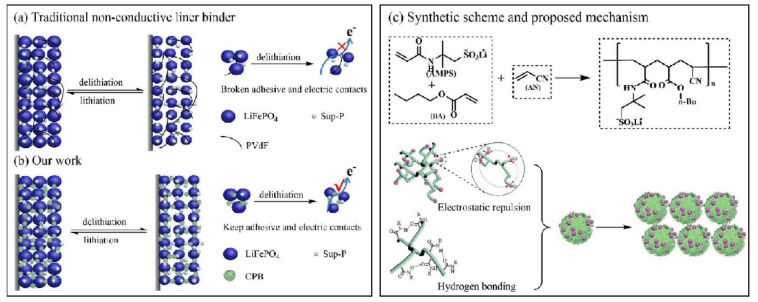
Schematics of the approaches to addressing electrode integrity issues in battery materials. (**a**) Traditional approaches using PVDF as a non-conductive polymer binder. (**b**) Replacing one-dimensional binders or two-dimensional binders, a 3D conductive binder could maintain the electrical and mechanical integrity of the electrode during charge/discharge cycles. (**c**) Schematic illustration of the synthesis and proposed mechanism of the conductive polymer binder (CPB) in this study. Reprinted with permission from ref. [79]. Copyright 2018 Elsevier.

**Table 1 polymers-13-03146-t001:** Summary of the main studies related to LiFePO_4_ cathodes obtained using various polymer binders.

Binder	Content of Binder (%)/Conductive Additive (%)/Solvent	Rate/Retention	Voltage Range (V)/Temperature (°C)	Ref./Year
WSB+CMC	4/6/water	1C/100% @200 cycles	2.5–4.0/60	[47]/2007
PAA	10/0/water	0.2 mA g^−1^/98.8% @50 cycles	2.5–4.2/25	[49]/2009
CMC	5/7/water	1C/75% @1000 cycles	2.8–4.2/25	[50]/2010
Lithiated perfluorosulfonate ionomer	20/20/NMP	-/-	2.5--/25	[92]/2011
Cellulose	4/10.7/water	1C/92% @50 cycles	2.8–4.2/25	[93]/2012
CMC-Li	10/10/water	-/96.2% @200 cycles	2.0–4.2/25	[69]/2014
PAA	7/10.3/NMP	175 mA g^−1^/99.3% @200 cycles	2.5–4.2/25	[94]/2012
PolyVIMTFSI-c	10/10/NMP	-/-	2.8–4.2/25	[84]/2013
PFSILi+PVDF	10/10/NMP	-/-	2.7–4.2/25	[65]/2013
PVDF-HFP	10/10/NMP	1C/- @120 cycles	2.5–4.2/25	[39]/2014
CMC-Li	10/10/water	-/-	2.0–4.2/25	[66,67]/2014
SA-PProDOT	20/0/water	1C/86.6% @400 cycles	2.0–4.2/25	[78]/2015
CN-CCTS	5/5/water	0.2C/99.2% @100 cycles	2.5–4.0/25	[52]/2015
PEDOT:PSS	4/4/water	1C/99% @150 cycles	2.0–4.0/25	[72]/2015
PTFE	5/5/water	0.2C/97.5% @100 cycles	2.0–3.8/25	[42]/2015
PEDOT:PSS+CMC	4/0/water	1C/99% @100 cycles	2.0–4.0/20 ± 2	[73]/2015
PEDOT:PSS	8/0/water	1C/100% @100 cycles	2.8–4.2/25	[74]/2015
polyurethane	10/10/water	-/-	2.8–4.2/25	[95]/2015
PVDF-TrFE	10/10/organic	1C/89% @50 cycles	2.5–4.2/25	[40]/2016
PEDOT:PSS	8/0/water	-/-	-	[75]/2016
CCTS+PEDOT:PSS	4/6/water	0.2C/100% @100 cycles	2.5–4.0/25	[76]/2016
Jeffamine^®^ compounds	30/7	0.1C/92% @ 100 cycles	2.4–3.9/70	[96]/2017
Xanthan gum (XG)	5/5	0.2C/96.9% @100 cycles	2.5–4.0/25	[97]/2017
L-spandex	10/10	0.5C/100.7% @100cycles	2.0–4.0/25	[98]/2017
PSBA-Li	1.5/3/water	0.5C/108.5% @200 cycles	2.5–3.7/25	[71]/2018
PAALi-TS	5/5/water	0.2C/97.47% @100 cycles	2.5–4.0/25	[70]/2018
SEBS	10/10/organic	2C/87% @50 cycles	2.5–4.2/25	[99]/2018
CPB	5/5/water	0.2C/94.5% @100 cycles	2.8–4.2/25	[79]/2018
CN-PEI	7/13/water	0.5C/99.6% @100 cycles	2.5–4.0/25	[53]/2018
Lithium sulfonate-grafted P(VDF-HFP)	10/10/NMP	1C/92%@50 cycles	2.4–4.3/25	[100]/2018
PEDOT:PSS-SPPO	5/0/water	C/3 /99.6% @ 30 cycles	2.0–4.0/25	[77]/2019
LPPS	10/10/water	1C/170 mAhg^−1^@ 200 cycles	2.5–4.2/25	[80]/2019
PVDF/TX	5/5/NMP	0.2C/ 97.4%@100 cycles	2.5–4.0/25	[101]/2019
PEGMA	10/5	0.25C/127 mAhg^−1^@ 200th cycle	2.0–4.0/25	[102]/2019
PVDF/Polyimide	10/10 (3:7)	1C/141 mAhg^−1^@ 300 cycles	2.5–3.7/25	[41]/2020
PPDI	8/2/organic	1C/nearly 100%@ 50 cycles	-	[103]/2020
SFPI	10/10/organic	0.5C/94.1% @100 cycles	2.5–3.8/25	[104]/2021

## Data Availability

The data presented in this study are available on request from the corresponding author.

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
