# Peer review of "Rational Design of Effective Binders for LiFePO4 Cathodes"

_polymers, 2021, doi:10.3390/polym13183146_

Round 1
Reviewer 1 Report
Shu Huang and coworkers review the current status and perspective of development of the polymer binder for LiFePO4 Cathodes. They discussed three class of binders: 1) conventional and commercialized PVDF and related composites, 2) waterborne binders, and 3) conductive binders. Their discussion is focused on the selection of functional groups and polymer chains. At the end of the review, the recent progress in this research area is summarized.
I think this review is a kind of preface review of a special issue of the journal, polymers, and the content may be appropriate. However, some parts of paper are not clear nor friendly to the readers, and would be improved for publication. My main concerns are listed below.
- Although the authors claimed that the mechanisms of binder structures in promoting Li-ion diffusion of LFP materials are highlighted to clarify the interaction and relationship between polymer structure, material particle surface, and electrochemical performance. However, no systematic categorization among polymer structure, material particle surface, and electrochemical performance is presented. The authors should not present just enumeration of previous works, but overview by sorting out the essential points of the works.
- Although the authors claimed that they forecast future binder research directions, as well as predict the development tendency of binder systems of LFP cathodes of LIBs and other rechargeable batteries by summarizing the recent progress. However, they just show the trend. The authors are advised to show a clear strategy from the viewpoint of key characteristics of binder such as adhesive function, structural stability and conductivity.
- Figures in the manuscript are not fully discussed. The authors are advised to rearrange the figures in order to be corresponded to the main text, or to discuss all the figures.
- On of miner concern is that the authors use Tg without definition.
From above reasons, the authors are advised to revise the manuscript for further consideration of publication in polymers.
Author Response
Response to Reviewer 1:
Comments:
Shu Huang and coworkers review the current status and perspective of development of the polymer binder for LiFePO4 Cathodes. They discussed three class of binders: 1) conventional and commercialized PVDF and related composites, 2) waterborne binders, and 3) conductive binders. Their discussion is focused on the selection of functional groups and polymer chains. At the end of the review, the recent progress in this research area is summarized.
I think this review is a kind of preface review of a special issue of the journal, polymers, and the content may be appropriate. However, some parts of paper are not clear nor friendly to the readers, and would be improved for publication. My main concerns are listed below.
- Although the authors claimed that the mechanisms of binder structures in promoting Li-ion diffusion of LFP materials are highlighted to clarify the interaction and relationship between polymer structure, material particle surface, and electrochemical performance. However, no systematic categorization among polymer structure, material particle surface, and electrochemical performance is presented. The authors should not present just enumeration of previous works, but overview by sorting out the essential points of the works.
Answer: Thanks for the reviewer’s valuable and helpful comments. We have added the related information in the manuscript as follows.
“Based on recent outcomes, plenty efforts have been focused on polymer structures and functions. For polymer structures, one approach is to design and prepare polymers with flexible component to form uniform electrodes with low roughness binder-coated surface films. Another approach is to decorate the polymer chains with functional groups, such as carboxyl, hydroxyl, and ester groups, which could promote the chemical bonding between LFP particles to enhance adhesion. Another vital approach is to introduce conductive structures. Therefore, the introduction of functional groups mainly play a role on forming hydrogen bond or strong interaction between LFP particle surfaces. Another function focus on promoting the diffusion of Li+ to shorten the pathway of Li+ to particle surfaces. The efforts could be to decrease the addition amount of binders to enhance energy densities and conductivity, and enhance the transformation of Li+ at the interface between the electrode and electrolyte to overcome the shortage of the LFP structure. So, the ultimate goals would be to make up the intrinsic drawbacks of olivine structure LiFePO4 to realize the long-term cycle life and excellent rate performances especially at high rates.”
- Although the authors claimed that they forecast future binder research directions, as well as predict the development tendency of binder systems of LFP cathodes of LIBs and other rechargeable batteries by summarizing the recent progress. However, they just show the trend. The authors are advised to show a clear strategy from the viewpoint of key characteristics of binder such as adhesive function, structural stability and conductivity.
Answer: Thanks for the reviewer’s valuable and helpful comments. We have added the related information in the manuscript as follows.
“Additionally, more attention should be paid to binders with multiple functions on one polymer chain. The mainly groups could contain: 1) hydrophilic groups to realize waterborne process; 2) -COOH, -OH or -NH2 groups which could form hydrogen bond with LFP particle surfaces with enhanced adhesion; 3) ion conductive of ionic groups to promote the diffusion of Li+; 4) crosslinked structures to maintain the structural stability of polymers upon cycling.”
- Figures in the manuscript are not fully discussed. The authors are advised to rearrange the figures in order to be corresponded to the main text, or to discuss all the figures.
Answer: Thanks for the reviewer’s valuable and helpful comments. We have rearrange a figure and added the related illustration in the manuscript as follows.
“Zhang et. al.[41] used polyimide (PI) (Figure 1a) to replace part of PVDF and the LFP-PI-0.7 (mixture of 30 wt.% PVDF and 70 wt.% PI as binders) electrodes yield a significantly cycling stability during 300 cycles at 1C with the stabilized capacity of 141 mA h g-1 (Figure 1b). Moreover, the capacity retentions of LFP-PI electrodes present a promotion with the increase of PI contents (Figure 1c). These outcomes owe to high ion conductivity of the PI binder, which is 3.6-fold higher than those of PVDF (Figure 1d). The PI binder did not only maintain stable distribution between LFP particles to effectively restrain the re-aggregation of LFP particles upon repeated cycling processes but also boosted ion diffusion through electrodes. Besides, the active carbonyl groups in PI skeleton may yield extra capacity due to the reversible enolization reaction.”
Figure
Figure 1. (a) Chain structures of PVDF and PI binders. (b) Cycling performances and (c) rate capabilities at 1C of LFP-PI-0.3, LFP-PI-0.5, and LFP-PI-0.7 electrodes. (d) Li-ion diffusion coefficients (DLi+) of LFP-PVDF and LFP-PI electrodes. Reproduced from Nanotechnol. Rev., Vol 9, Zhang, Q.; Sha, Z. F.; Cui, X.; Qiu, S. Q.; He, C. G.; Zhang, J. L.; Wang, X. G.; Yang, Y. K. Incorporation of redox-active polyimide binder into LiFePO4 cathode for high-rate electrochemical energy storage. pp. 1350-1358, Copyright (2020), with permission from Walter de Gruyter GmbH.
We have also added the discussion of all the figures in the manuscript. For example:
“Novel binder systems soluble in water and compatible with cathodes have also attracted increasing attention. For instance, a novel water-soluble elastomer binder (WSB)[47] was added to carboxymethyl cellulose (CMC) to enhance flexibility and adhesion of binders in elastomer-based LiFePO4 cathode (Figure 3a and b). The improved flexibility could be beneficial for enhancing the energy density of electrodes. Moreover, the SEM images present uniform electrodes with low roughness WSB-coated surface films (Figure 3c and d). By the evaluation of the electrode flexibility (Figure 3e), the results apparently show that the elastomer system is much more flexible than the PVdF systems, which is about 2-fold higher (Figure 3f). The resulting cathodes showed good performance with capacity reaching 120mAh g-1 at 10 C and 60 oC (Figure 3g).”
“Lux et.al.[50] introduced CMC aqueous binder alone to yield LFP cathodes with an initial capacity of 140 mAh g-1 and capacity fading of only 0.025% per cycle. Sun et. al.[51] used water-soluble chitosan derivative (carboxymethyl chitosan (C-CTS)) to investigate CMC-relevant structures (Figure 4a). The obtained LFP electrodes were more resistant to the electrolyte and showed high rate performances than those prepared with PVDF and CMC. Furthermore, introducing cyanoethyl group into C-CTS chains to prepare cyanoethylated carboxymethyl chitosan (CN-CCTS) polymer could yield highly adhesive binders (Figure 4 b and c). The adhesion strength of CN-CCTS/LFP cathodes was 3.6-fold higher than that of CMC. Cyclic voltammetry (CV) and electrochemical impedance spectroscopy (EIS) (Figure 4 e) results revealed CN-CCTS/LFP cathodes with favorable electrochemical kinetics than those prepared with CMC and PVDF, thereby better cycling performances (Figure 4 d).”
“In this respect, sodium alginate (SA) was functionalized with 3,4-propylenedioxythiophene-2,5-dicarboxylic acid (ProDOT) (Figure 6a) to form SA-PProDOT polymer chains[78] containing carboxyl, hydroxyl, and ester groups that could promote the chemical bonding between LFP particles. The resulting material showed stronger adhesion (Figure 6b). SA and ProDOT molecules may self-assemble through hydrophilicity of functional groups to yield interfaces for esterification reactions (Figure 6c). In this process, produced water could be removed from the interface to the hydrophilic phase, promoting the reaction equilibrium forward. The attachment of ProDOT groups could enhance the polarity of the polymer, as well as its uptake of the electrolyte, thereby promoting the ionic conductivity (Figure 6c). Without conductive additives, the fabricated cathodes could present enhanced rate performances (Figure 6d) and achieve theoretical specific capacities of LiFePO4 cathodes, estimated to 170 mAh g-1 at C/10 and 120 mAh g-1 at 1C over 400 cycles (Figure 6e).”
- On of miner concern is that the authors use Tg without definition.
From above reasons, the authors are advised to revise the manuscript for further consideration of publication in polymers.
Answer: Thanks for the reviewer’s valuable and helpful comments. We have added the related information in the manuscript as follows.
“For instance, Hu et al. [39] used poly(vinylidenefluoride-co-hexa-fluoropropylene) (PVDF-HFP) binder in LFP cathodes to yield good electrochemical performances due to its elevated amorphous and electronic conductivity coupled with lower glass transition temperature (Tg).”

Reviewer 2 Report
I have read the review article titled “Rational design of effective binders for LiFePO4 Cathodes” by Chen et al submitted to Polymers. The discussed subject content on polymer binders and their effect on well-known lithium iron phosphate (LFP) cathode material is suitable for this journal.
The lithium-ion battery is an attractive energy storage device exhibiting excellent efficiency of charge-discharge processes along with high energy density among the available other rechargeable systems. They are an indispensable power source for a variety of portable electronic devices today. Conventional lithium-ion batteries (invented by Sony) have graphite as anode and lithium transition (Cobalt) metal oxides as the cathode in the battery system. In the last decade, to further enhance the energy density of the storage system, extensive research has been undertaken towards developing new electrodes as the cathode. In this, olivine compounds are an excellent material to act as a cathode in a lithium battery system. LFP is well known and suitable for EVs. Over the past decade, the design of binder molecules has gone through tremendous evolution from primarily maintaining the structural integrity of the electrode against volume change and facilitate ion transport in the charge and discharge processes.
This article reviewed the development of binder from the perspective of molecular design, and comprehensively discusses the correlation between the functions of the binder molecules and the cell performance.
Hence, the submitted review work is significantly providing a comprehensive review of the progress in the design and function of the binder (including alginate, traditional PVDF, etc.). The review is well presented and well written with supporting characterizations. However, some revision is required before rendering a final decision.
- In section 1; The principles and challenges of LFP cathodes in terms of conductivity and how binders can mitigate these issues can be explained.
- The operation principle of the LFP system can be briefly explained in this section.
- Section 1; lines 81 -82 Are PTFE also used in the LFP system (or) only PVDF?
- Dr. Manickam Minakshi and his group have worked extensively worked on rational designing of olivine materials with an emphasis on binders (like PVDF, PTFE, Alginate, Chitosan, etc.), please include and discuss.
- Line 116; what is the chemical structure of the PI binder?
- Please make sure that all the figures in Fig. 2 are explained in the text.
- Are the waterborne binders accommodate the stress/strain (structural/mechanical stability) during the electrochemical process?
- How about PVP, PVA, and PEI linear polymers?
- What would be the optimum weight percent of binder to be included in the LFP cathode?
- Are the conductive binders act as intercalation hosts?
- Performance comparisons of representative binders with specific functions can be included in the conclusions and perspective section.
- The stated binders will be also suitable for the NaFePO4 system?
Author Response
Response to Reviewer 2:
Comments
I have read the review article titled “Rational design of effective binders for LiFePO4 Cathodes” by Chen et al submitted to Polymers. The discussed subject content on polymer binders and their effect on well-known lithium iron phosphate (LFP) cathode material is suitable for this journal.
The lithium-ion battery is an attractive energy storage device exhibiting excellent efficiency of charge-discharge processes along with high energy density among the available other rechargeable systems. They are an indispensable power source for a variety of portable electronic devices today. Conventional lithium-ion batteries (invented by Sony) have graphite as anode and lithium transition (Cobalt) metal oxides as the cathode in the battery system. In the last decade, to further enhance the energy density of the storage system, extensive research has been undertaken towards developing new electrodes as the cathode. In this, olivine compounds are an excellent material to act as a cathode in a lithium battery system. LFP is well known and suitable for EVs. Over the past decade, the design of binder molecules has gone through tremendous evolution from primarily maintaining the structural integrity of the electrode against volume change and facilitate ion transport in the charge and discharge processes.
This article reviewed the development of binder from the perspective of molecular design, and comprehensively discusses the correlation between the functions of the binder molecules and the cell performance.
Hence, the submitted review work is significantly providing a comprehensive review of the progress in the design and function of the binder (including alginate, traditional PVDF, etc.). The review is well presented and well written with supporting characterizations. However, some revision is required before rendering a final decision.
- In section 1; The principles and challenges of LFP cathodes in terms of conductivity and how binders can mitigate these issues can be explained.
Answer: Thanks for the reviewer’s valuable and helpful comments. We have added the related information in the manuscript as follows.
“In the process of slurry preparation, the polymer chains would coat on LFP particle surfaces upon stirring. Then the produced slurry would be coated on aluminum foils and electrodes would be made after drying. Therefore, the polymer structure coated on the LFP particle surface would significantly affected performances of electrodes. The efficient polymer structures could promote the diffusion of Li+ in a degree, which would be helpful to mitigate the shortage of one-dimensional Li+ channels of LFP material.”
- The operation principle of the LFP system can be briefly explained in this section.
Answer: Thanks for the reviewer’s valuable and helpful comments. We have added the related information in the manuscript as follows.
“In the process of slurry preparation, the polymer chains would coat on LFP particle surfaces upon stirring. Then the produced slurry would be coated on aluminum foils and electrodes would be made after drying. Therefore, the polymer structure coated on the LFP particle surface would significantly affected performances of electrodes. The efficient polymer structures could promote the diffusion of Li+ in a degree, which would be helpful to mitigate the shortage of one-dimensional Li+ channels of LFP material.”
- Section 1; lines 81 -82 Are PTFE also used in the LFP system (or) only PVDF?
Answer: Thanks for the reviewer’s valuable and helpful comments. The PVDF binder need to use highly toxic organic NMP as solvents. Therefore, waterborne fluorine-containing polymer polytetrafluoroethylene (PTFE) was prepared and used to replace organic PVDF in LiFePO4/C cathodes. The PTFE used as the binder in the LFP system to compared with the LFP system with PVDF only.
- Manickam Minakshi and his group have worked extensively worked on rational designing of olivine materials with an emphasis on binders (like PVDF, PTFE, Alginate, Chitosan, etc.), please include and discuss.
Answer: Thanks for the reviewer’s valuable and helpful comments. We have added the related information in the manuscript as follows.
“In addition, these binder system might also suitable for other olivine materials, such as olivine LiNiPO4,[88] LiCo1/3Mn1/3Ni1/3PO4,[89] and LiCoPO4 [90] cathodes, with the traditional PVDF binder. Moreover, the introduction of chitosan cross-linked by electrodeposit method for the Zn-MnO2 system with excellent electrochemical properties, and which would be a promising strategy for further researches.[91]”
[88] Minakshi, M.; Singh, P.; Appadoo, D.; Martin, D. E. Synthesis and characterization of olivine LiNiPO4 for aqueous rechargeable battery. Electrochim. Acta. 2011, 56, 4356-4360.
[89] Kandhasamy, S.; Singh, P.; Thurgate, S.; Ionescu, M.; Appadoo, D.; Minakshi, M. Olivine-type cathode for rechargeable batteries: Role of chelating agents. Electrochim. Acta. 2012, 82, 302-308.
[90] Minakshi, M.; Kandhasamy, S. Influence of sol-gel derived lithium cobalt phosphate in alkaline rechargeable battery. J Sol-Gel Sci Technol. 2012, 64, 47-53.
[91]Biswal, A.; Minakshi, M.; Tripathy, B. C. Probing the electrochemical properties of biopolymer modified EMD nanoflakes through electrodeposition for high performance alkaline batteries. Dalton Trans., 2016, 45, 5557-5567.
- Line 116; what is the chemical structure of the PI binder?
Answer: Thanks for the reviewer’s valuable and helpful comments. We have added the chemical structure of the PI binder in the manuscript as follows.
Figure
Figure 1. (a) Chain structures of PVDF and PI binders. (b) Cycling performances and (c) rate capabilities at 1C of LFP-PI-0.3, LFP-PI-0.5, and LFP-PI-0.7 electrodes. (d) Li-ion diffusion coefficients (DLi +) of LFP-PVDF and LFP-PI electrodes.
- Please make sure that all the figures in Fig. 2 are explained in the text.
Answer: Thanks for the reviewer’s valuable and helpful comments. We have added the explanation in the manuscript as follows.
“Novel binder systems soluble in water and compatible with cathodes have also attracted increasing attention. For instance, a novel water-soluble elastomer binder (WSB)[47] was added to carboxymethyl cellulose (CMC) to enhance flexibility and adhesion of binders in elastomer-based LiFePO4 cathode (Figure 3a and b). The improved flexibility could be beneficial for enhancing the energy density of electrodes. Moreover, the SEM images present uniform electrodes with low roughness WSB-coated surface films (Figure 3c and d). By the evaluation of the electrode flexibility (Figure 3e), the results apparently show that the elastomer system is much more flexible than the PVdF systems, which is about 2-fold higher (Figure 3f). The resulting cathodes showed good performance with capacity reaching 120mAh g-1 at 10 C and 60 oC (Figure 3g).”
- Are the waterborne binders accommodate the stress/strain (structural/mechanical stability) during the electrochemical process?
Answer: Thanks for the reviewer’s valuable and helpful comments. Based on the researches, waterborne binders could accommodate the stress/strain (structural/mechanical stability) during the electrochemical process. For example, researches of references 49, 51, 52 and 53 have tested surface and cross-sectional morphology of electrodes fabricated with waterborne binders before and after cycling. And the SEM images (Figure 1 and 2) show that cathodes present no obvious morphological change and cracks and demonstrate that waterborne binders could accommodate the stress/strain and keep structural/mechanical stability during cycling.
Figure
Figure 1. SEM images of LFP electrodes with C-CTS, (a,c) before and (b,d) after 200 cycles.[51]
Figure
Figure 2. SEM images of LFP electrodes with different binders, (a, c, e) before and (b, d, f) after 100-cycle charge–discharge test: a,b CN-PEI-7, c, d PVDF and e, f PEI binder, respectively.[53]
[51] Sun, M. H.; Zhong, H. X.; Jiao, S. R.; Shao, H. Q.; Zhang, L. Z. Investigation on Carboxymethyl Chitosan as New Water Soluble Binder for LiFePO4 Cathode in Li-Ion Batteries. Electrochim. Acta 2014, 127, 239-244.
[53] Huang, J.; Wang, J.; Zhong, H.; Zhang, L. N-cyanoethyl polyethylenimine as a water-soluble binder for LiFePO4 cathode in lithium-ion batteries. J. Mater. Sci. 2018, 53, 9690-9700.
And we have added the related figures and description in the manuscript as follows.
Figure
Figure 4. (a) The synthetic route of CN-CCTS. Photographs of the LFP electrode sheets with CN-CCTS (b) and CCTS (c) after folding for three times.(d) Cycling performance of LFP electrode with CN-CCTS, CMC and PVDF binder. (e) Nyquist plots of LFP electrode with CN-CCTS, CMC and PVDF binder. SEM images of LFP electrodes with the CN-CCTS binder, (f) before and (g) after 100 cycles charge-discharge test. Reproduced from Electrochim. Acta, Vol 182, He, J. R.; Wang, J. L.; Zhong, H. X.; Ding, J. N.; Zhang, L. Z. Cyanoethylated Carboxymethyl Chitosan as Water Soluble Binder with Enhanced Adhesion Capability and electrochemical performances for LiFePO4 Cathode. pp. 900-907, Copyright (2015), with permission from Elsevier.
“After 100 cycles, the SEM images of CN-CCTS/LFP cathode show no obvious morphological change and cracks (Figure 4 f and g). The result demonstrates that the waterborne CN-CCTS binder could accommodate the stress/strain and keep structural/mechanical stability during cycling.”
- How about PVP, PVA, and PEI linear polymers?
Answer: Thanks for the reviewer’s valuable and helpful comments. These linear polymers might be suitable for LFP cathodes, which could be verified for further researches. Based on recent outcomes shown in Table 1 in the manuscript, the conductive polymer, ionomers, and ionic polymers would be more suitable for LFP cathodes for its intrinsic drawbacks of olivine structure. Therefore, the PVP, PVA and PEI polymers modified with conductive or ionic groups might be helpful for enhancing performances, and which might become a promising strategy and inspiration for further researches.
- What would be the optimum weight percent of binder to be included in the LFP cathode?
Answer: Thanks for the reviewer’s valuable and helpful comments. The addition amount of PVDF for LFP cathodes is about 5 wt.% in commercial application. Based on recent researches shown in Table 1 in the manuscript, the optimum weight percent of a binder for LFP cathodes would be below 5 wt.% to maintain the adhesion and energy densities of electrodes. Moreover, this content is also affected by the type of binders. For example, if a binder is a kind of conductive polymer, which could replace the whole or part of conductive additives, the content of the binder might be higher.
In addition, we have added the prediction in the manuscript as follows.
“The addition of binders could be decreased to below 5 wt.% for further application, beneficial to increasing the energy densities of electrodes.”
- Are the conductive binders act as intercalation hosts?
Answer: Thanks for the reviewer’s valuable and helpful comments. Based on outcomes summarized in Table 1 in the manuscript. The conductive binders have attracted main attentions and presented excellent performances. Therefore, conductive binders could be acted as intercalation hosts.
- Performance comparisons of representative binders with specific functions can be included in the conclusions and perspective section.
Answer: Thanks for the reviewer’s valuable and helpful comments. We have added the explanation in the manuscript as follows.
“The fundamental design principles can be summarized by several points. The first consisted of a waterborne process advantageous in terms of eco-environment, low-cost, and non-toxic solvent. The second dealt with enhancing conductive ability. The high conductive PEDOT:PSS could replace conductive additives to fabricate cathodes. And the third facilitating the transformation of Li+. The lithiated ionomer (PSBA-Li) binder could promote the transformation of Li+ in electrodes and shorten the distance of Li+ from the electrolyte to the surfaces of LiFePO4 particles, leading to enhanced cycling stability. The fourth point had to do with strengthening the adhesion ability with less added amounts of binders. SA-PProDOT polymer chains with carboxyl, hydroxyl, and ester groups could promote the chemical bonding between LFP particles, leading to stronger adhesion. Moreover, PSBA-Li binder could be used with a limited amount of 1.5 wt.% to fabricate LFP cathodes for the function of -COOH. The fifth relied on enhanced functionalized structures with various functional groups on polymer chains.”
- The stated binders will be also suitable for the NaFePO4system?
Answer: Thanks for the reviewer’s valuable and helpful comments. The NaFePO4 system has the similar principle compared to the LiFePO4 system. Therefore, traditional binders, such as PVDF, are suitable for the NaFePO4 system. The waterborne process of the NaFePO4 system would rely on the modification method of material surfaces. Many conductive binders might be suitable for the NaFePO4 system with enhanced performances. However, the lithiated ionomers might be unsuitable for the NaFePO4 system, because Li+ provided by lithiated structures could not promote the diffusion of Na+ to enhance performances. The ionic polymers, such as ionic-liquid-based copolymers, might be suitable for the NaFePO4 system, which need further researches.

Round 2
Reviewer 1 Report
Shu Huang and coworkers review the current status and perspective of development of the polymer binder for LiFePO4 Cathodes. They discussed three class of binders: 1) conventional and commercialized PVDF and related composites, 2) waterborne binders, and 3) conductive binders. Their discussion is focused on the selection of functional groups and polymer chains. At the end of the review, the recent progress in this research area is summarized.
I think this review is a kind of preface review of a special issue of the journal, polymers, and the content may be appropriate. The resubmitted manuscript has been revised according to the referees’ suggestions. Therefore, I recommend the manuscript to be published in polymers.
Reviewer 2 Report
The authors have responded to my queries. To my opinion, the revised version is suitable to publish